# Usefulness of Discriminant Analysis in the Morphometric Differentiation of Six Native Freshwater Species from Ecuador

**DOI:** 10.3390/ani11010111

**Published:** 2021-01-07

**Authors:** Ana Gonzalez-Martinez, Carmen De-Pablos-Heredero, Martin González, Jorge Rodriguez, Cecilio Barba, Antón García

**Affiliations:** 1Department of Animal Production, Faculty of Veterinary Sciences, University of Cordoba, 14071 Córdoba, Spain; agmartinez@uco.es (A.G.-M.); cjbarba@uco.es (C.B.); 2Department of Business Economics (Administration, Management and Organization), Applied Economics II and Fundamentals of Economic Analysis, ESIC Business & Marketing School, Rey Juan Carlos University, Paseo de los Artilleros s/n, 28032 Madrid, Spain; carmen.depablos@urjc.es; 3Department of Animal Production, Quevedo State Technical University, Av. Quito km. 1 1/2 vía a Santo Domingo de los Tsáchilas, Quevedo, 120501 Los Ríos, Ecuador; mgonzalez@uteq.edu.ec (M.G.); jrodriguez@uteq.edu.ec (J.R.)

**Keywords:** fish, Guayas River Basin, morphostructure, multivariate techniques

## Abstract

**Simple Summary:**

Ecuador is among the 25 highest fish producers in the world and the fishing sector contributes 7% to the national consumption of animal proteins. This country has a high number of native freshwater species, some of which are endangered due to the modification, fragmentation and destruction of habitats; introduction of foreign species; overfishing with illegal gear; environmental pollution; development of large-scale intensive forestry practices; loss of ecological continuity between different rivers; and climate change. The Guayas, the largest basin (CHG) in the Pacific Ocean located in Ecuador, constitutes an important biodiversity reserve of native freshwater. Most part of these 125 species are endemic and threatened or in an at-risk situation. It would be useful to analyze the morphological differences among six existing species in the CHG, and to select a reduced number of direct, simple and low-cost measurements to be applied in marginal communities. In this work, the usefulness of discriminant analysis was proven in the differentiation of six native freshwater species. Results represent a new step in the development of breeding and conservation plans for this native zoogenetic resource.

**Abstract:**

The aim of this research was to find out the morphometric differentiation of six native freshwater species in the Guayas Hydrographic Basin (Ecuador) by means of discriminant analysis. A total of 1355 mature fishes (*Cichlasoma festae, Andinoacara rivulatus, Dormitator latifrons, Bryncon dentex, Hoplias microlepis* and *Leporinus ecuadorensis*) were captured and 27 morphometric measurements and 20 landmarks were used. Two-way analysis of variance with species and sex as fixed factors and discriminant analysis were applied. The selection of the most discriminant variables was made applying the F of Snedecor, Wilks’-Lambda and the 1-Tolerance. While sex within species had no significant effect on the morphology, differences among species were significant. Twenty-seven morphological variables showed highly significant differences among six native freshwater species. Nine biometric variables with high discriminant power were selected. The six species analyzed were discriminated by the morphometric models generated, thus showing that discriminant analysis was useful for differentiating species. The morphometric differentiation by discriminant analysis is a direct, simple and economic methodology to be applied in situ in rural communities. It favors the implementation of a livestock development program and it could be used with other native freshwater species in the Guayas Hydrographic Basin.

## 1. Introduction

Ecuador is among the top 25 fish producers in the world [1]. The Fisheries and Fisheries Development Law from Ecuador [2] establishes that bioaquatic resources (of the territorial sea, inland maritime waters, rivers, lakes or natural and artificial channels) are national assets, with the State in charge of regulating their rational use (extraction, cultivation, processing and commercialization). In addition, the State is also in charge of promoting the investigation of bioaquatic resources, promoting the creation of educational centers for the training and qualification of the personnel necessary for the fishing activity and designing the required promotion measures for the expansion of the fishing sector.

The Ecuadorian fishing sector contributes 7% to the national consumption of animal proteins (5–8 kg/year/head) [3]. Inland fishing represents around 338 t [3], being an important contribution of food to the populations located near of capture areas. In addition to the catches made by fishermen through artisanal practices, Ecuador stands out for having the highest small-scale artisanal fisheries, counting on with responsible practices and environmentally sustainable fishing activities that promote the conservation of aquaculture resources and their ecosystems [4]. This aquaculture can contribute to food security in the poorest rural areas, promote social cohesion and local endogenous development, while improving livelihoods and the conservation of biodiversity [5]. Ecuadorian aquaculture, despite a series of limitations as lack or limited amount of seed from cultivated organisms and little technical assistance [6], presents favorable conditions to be able to produce products throughout the year [7].

The river network of Ecuador is complex and diverse, highlighting the Guayas River Basin (CHG), with an extension of 53,299 km^2^ (Figure 1), as the largest basin from South America on the Pacific side of the continent [8]. It extends between the parallels 00°14′ S and 02°27′ S, and the meridians 78°36′ W and 80°36′ W. The CHG is made up of seven sub-basins with high geomorphological, climatic and biological diversity [9]. It is a subsidence pit with marine fluvial filling configured to the east by the Andes Mountain, to the west by the Chongón-Colonche Coast Mountain, to the north by the sedimentary reliefs of the Esmeralda river and to the south by the Guayas river delta, consequence of the confluence of Daule and Babahoyo rivers [10,11].

The CHG shows great aquaculture diversity, with an inventory of 123 species belonging to 14 orders [12]. A large part of these species are local aquaculture endemics adapted to different habitats (freshwater, estuaries and marine), configured by different internal and external factors, among which the quality of the water, the type of soil and the vegetation stand out [9]. Fishes live in a fragile balance in the aquatic environment, being very vulnerable to stress and diseases [13,14].

The first studies of the continental ichthyological richness of Ecuador began in the 19th century, with the first list of fish from Ecuador appearing in 1968 (Ovchynnick) and it included a total of 295 species, increased to 306 species in a later study (1971). The list was increased to 706 and 824 species by Barriga’s studies in 1991 and 2012, respectively [9,15]. However, knowledge of the diversity, distribution and ecology of freshwater ichthyofauna presents numerous gaps.

The Food and Agriculture Organization (FAO) [16] considered it a priority to do species and breeds characterization studies as the first phase in implementation of a livestock development program. Research on the native species and breeds of Latin America has found that the main problem was lack of data and an absence of characterization and productive behavior studies [17]. Characterization of animal genetic resources covers all activities associated with the identification, and quantitative and qualitative description of populations, and the natural habitat and production systems to which they are adapted [18]. In this sense, morphologic analysis has been widely used for breed and characterization of populations. It has recently been used in comparative morphometric studies of creole cattle breeds in Argentina, Ecuador and Africa [19,20]; horses [21]; sheep [22]; goats [23]; pigs [24]; dogs [25]; ducks [26]; turkeys [27]; and fishes [28]. Species identification is a fundamental step towards any research work, being morphometric measurements and meristic counts of great utility for their identification and taxonomic study [6]. In general, fishes show greater variation and plasticity in morphological traits within and between populations than other vertebrates [29].

In fishes, morphometric analysis, due to the relative growth of the different body regions and components, is an important key in the study of their biology. Morphometric relationships between the various parts of the body by, for example, Fulton conduction’s factor can be used to assess the welfare of individuals and to determine the possible difference between separate unit stocks of the same species [30]. In this sense, fishes are very sensitive to environmental changes and adapt quickly when the morphometry change [29].

Previously, this research group has made the morphological, meat quality and productive characterization of different native freshwater species of Guayas River Basin located in the coast zone of Ecuador: *Cichlasoma festae* by González et al. [31,32] and Rodriguez et al. [4,8]; *Andinoacara rivulatus* by Caez et al. [33] and González et al. [34]; *Dormitator latifrons* by González-Martínez et al. [35]; *Hoplias microlepis* by González-Martínez et al. [36]. The situation of these native endemic fish water populations in Ecuador includes risk criteria to consider them as threatened species, including the reduction of population; modification, fragmentation and destruction of habitats; the increase of non-native species (*Oreochromis* spp.); overfishing with illegal gear; environmental contamination; loss of modification of natural hydrological regime, including basin river connectivity; and, finally, climate change [8,37,38,39]. In each case, factors such as sex, production system (wild and cultured) and capture area, among others, were studied. However, we considered that, independently of endogenous factors and basin habitat modification, there must be a high homogeneity within each species and a highlighted distance amongst species. It would be of great interest to analyze the morphological differences among the existing species in the CHG, subjected to similar modifications in the ecosystem and to assess the distances of both closer and far away species.

Morphometric characteristics have usually been used to differentiate species [40] or populations within the same species through multifactorial discriminant analysis [31,33,35,41]. These analyses provide a solid basis for the identification, rational management, breeding and conservation of fish genetic resources [42].

This work aims to study the usefulness of the discriminant analysis in the differentiation of six fish species from the Guayas Hydrographic Basin of Ecuador by morphometric measures. This methodology was used to differentiate species from each other and to assess the homogeneity within each species. It was also proposed as a direct methodology and appropriate for the objectives. This is the first time that morphological differentiation is made amongst freshwater species in Ecuador and results may be very useful to differentiate other existing species.

## 2. Materials and Methods

### 2.1. Data Collection

The study was carried out in the fluvial system of the Guayas Hydrographic Basin (CHG), within the province of Los Ríos and Manabi (Ecuador), which is characterized by a tropical climate with an average temperature of 25 °C, an annual rainfall of 2400 and 838.7 mm and a relative humidity of 82% and 78%, respectively [31,35].

A collection formed by 1355 mature fishes from *Cichlasoma festae* (N = 100; 48 females and 52 males), *Andinoacara rivulatus* (N = 300; 150 females and 150 males), *Dormitator latifrons* (N = 300; 147 females and 153 males), *Brycon dentex* (N = 214; 130 females and 84 males), *Hoplias microlepis* (N = 225; 132 females and 93 males) and *Leporinus ecuadorensis* (N = 216; 94 females and 122 males) (Figure 1) were captured in the last five years, both in rainy and dry season, although most of them were captured, after spawning, at the beginning of the dry season, between March and April. The fishes were caught using traditional fishing system called “wing deployment”, where the fishnet is swept in the opposite direction to the flow of water. The captured specimens were deposited in a cage attached to the net and were placed in a transition pond under natural conditions to transport them alive. The non-representative specimens of the species (less of 3% of the sample), with small size, fish of other species, mutilated, as well as young individuals were immediately returned to the river.

In the research center, the captured specimens were kept in two tanks with a capacity of 500 L each (dissolved oxygen = 6.20 ± 0.0 mg/L, temperature = 20.5 ± 0.2 °C and pH = 5.6 ± 0.1). The fishes remained in these tanks for 48 h, with a fasting time of 24 h before their death. After that, they were stunned by immersion in a mixture of water and ice at 0.8 °C for 20 min. Once the death was certified, the fishes were labeled and weighed with an electronic balance with a 0.1 g precision. Then, the morphometric measurements were obtained. Sex was determined through external phenotypic characteristics. The procedure was carried out according to the Canadian Council on Animal Care (CCAC) guide for investigative fish management [43]. We have tried to take all the necessary steps to assure the welfare of native fishes in this research, during the handling, transport and sacrifice phases. Nevertheless, the suggested method of euthanasia by the referenced authority (CCAC) was not used. According to the Guide to Good Practice for Slaughter [44], they were sacrificed. The animal welfare in each research step was supervised by a veterinary practitioner.

### 2.2. Body Measuerements

Morphometric measurements according to Diodatti et al. [45] were collected by the same person on the left side of each of the specimens under study, except those relative with widths and perimeters. An ichthyometer, tape measure and digital caliper graduated in millimeters were used. A total of 27 morphometric measurements were obtained, using 20 landmarks (Table 1, [35]), in agreement with the methodology used for native species of Ecuador [31,33,35].

### 2.3. Fulton’s Condition Factor (K)

Fulton´s condition factor (K) is a fish welfare index that allows obtaining useful information on growth, age, reproduction, nutrition, health and welfare status [31,33,46]. It was calculated using the equation K = (100 × BW)/SL^3^, where BW is body weight (g) and SL is standard length (cm).

### 2.4. Statistical Analysis

Statistical 12.0 for Windows software was used to perform the statistical analyses. Prior to the statistical analyses, the normality of the data distribution was evaluated using the Kolmogorov-Smirnov test (with the Lilliefords correction). For those variables that did not show a normal distribution, the Bartlett test was applied to assess if the data had equal variances.

The total length of *Brycon dentex* and *Leporinus ecuadorensis* specimens, due to their bifurcated caudal fin (Figure 1), was obtained by arithmetic mean of both total lengths. To avoid the possible biases produced by the effect of size in the morphometric variables, all of them were standardized following Elliot et al.’s [47] methodology, which was M_adj_ = M (L_s/_L_o_)^b^, where M is the original value of the morphometric measure, M_adj_ is the adjusted size of the measure, L_o_ the standard length of the fish and L_s_ the mean of the standard length of all fish. The parameter b was estimated for each character from the observed data of the regression curve of log M in log L_o_, using all fishes. This method normalizes the individuals of a species with a unique arbitrary size, common to all samples and, at the same time, maintaining the individual variation [48]. This adjustment has been used successfully by many authors [31,33,34,49,50]. The efficiency of the transformation was evaluated by testing the significance of the correlation between the transformed variable and the standard length of the fish.

The morphometric variables were compared using the analysis of variance (ANOVA), establishing the species (*Cichlasoma festae, Andinoacara rivulatus, Dormitator latifrons, Bryncon dentex, Hoplias microlepis* and *Leporinus ecuadorensis*) and sex as a fixed effect, with five and one degree of freedom for each factor, respectively. For the comparison of means, the Tukey method was used. A discriminant analysis was performed with all the transformed variables (males and females), including the classification matrix, as well as the graphic representation of the Mahanalobis distances through clusters and the spatial distribution of fishes through canonical Scatterplot, establishing the fish species as the grouping variable. A direct method of selection of variables was used at *p*-value < 0.05. Second, the selection of the most discriminant variables was made applying the F of Snedecor, Wilks’ Lambda and the 1-Tolerance. High values of F for each variable indicate that the means of each group are widely separated and these groups are homogeneous. Small Lambda values indicate that the variable discriminates well amongst groups. Finally, variables with a high percentage of tolerance (1-Toler) that reduced the redundant information were searched.

## 3. Results

### 3.1. Morphometric Characteristics

Sex had no significant effect (*p* > 0.05) on the morphology of the fish and was therefore not considered for displaying the rest of the results, being combined data from both sexes for the following analysis. Weight and total length differed significantly (*p* < 0.001) between species, with the mean values of these variables between 89.20 ± 3.01 g and 18.30 ± 0.27 cm (*Cichlasoma festae*) and 442.66 ± 29.52 g and 37.82 ± 0.69 cm (*Hoplias microlepis*), respectively (Table 2). The highest coefficients of variation were found in *Hoplias microlepis*, while *Cichlasoma festae* was the species with the highest homogeneity whose coefficients of variation did not reach 10% in a high number of variables.

The highest values in the ratio from body weight and morphometric variable with standard length of the fishes were found in *Cichlasoma festae*, *Andinoacara rivulatus* and *Bryncon dentex* (Appendix A), with *Leporinus ecuadorensis* being the species with the smallest values. All of these ratios were statistically significant (*p* < 0.001) for the whole sample, finding similarities among some species.

### 3.2. Morphological Differentation between Species

In order to establish relationships or differences between the six fish species considered, the morphometric measurements were first standardized, eliminating the effect of fish size. The standardization of the sample was effective, since all variables, except for body weight, condition factor, dorsal fin length, dorsal fin ray length and body width 3, were not correlated with SL. The variables adjusted for each of the species showed highly significant differences (*p* < 0.001) between the six species (Appendix A). The higher values in majority ratios were obtained in both *Cichlasoma festae* and *Andinoacara rivulatus,* and the lower in *Hoplias microlepis* and *Leporinus ecuadorensis*.

All morphometric adjusted variables included in discriminant analysis presented discrimination power, being in all cases significant (*p* < 0.001), except for body perimeter 2 (*p* > 0.05) (Table 3). According to F-Snedecor, Wilks’ Lambda and 1-Toler, the most discriminant variables in the model were: pre-dorsal fin length (Pre-DL), pre-orbital length (Pre-OL), pectoral fin length (PcFL), body depth 1 (AC1), pelvic fin length (PvFL), body depth 2 (AC2), body width 2 (LC2), body perimeter 3 (P3) and body width 1 (LC1).

The classification matrix resulted in correct ascription percentages of practically 100% (98.97), obtaining assignment errors only in *Bryncon dentex* (6) and *Hoplias microlepis* (1) (data no presented). The morphostructural differences of the six analyzed fish species are visually shown in Figure 2 and Figure 3. In the first one, in which the Mahalanobis distances obtained from the morphometric measurements are graphically represented, a first cluster grouped *Andinoacara rivulatus* and *Dormitator latifrons* specimens, a second cluster made up of *Hoplias microlepis* and *Leporinus ecuadorensis* specimens and a third cluster that includes *Cichlasoma festae*, *Andinoacara rivulatus*, *Dormitator latifrons, Hoplias microlepis* and *Leporinus ecuadorensis* species. The fishes from *Bryncon dentex* showed greater separation, and, therefore, greater morphostructural differentiation. The existence of different and unique morphostructural models for each species is observed in Figure 3, which shows a clear spatial distribution of each fish species, with little overlap of certain individuals of some of them. *Leporinus ecuadorensis*, *Andinoacara rivulatus* and *Bryncon dentex* showed the greatest homogeneity.

### 3.3. Fulton’s Condition Factor (K)

The mean value of Fulton’s condition factor (K) was 3.24 ± 0.08, obtaining the highest value in the *Andinoacara rivulatus* (5.40) and the lowest in *Hoplias microlepis* (1.47) (Table 2). The difference of K factor was highly significant (*p* < 0.001) especially among *Andinoacara rivulatus* and *Bryncon dentex* with other species, and the coefficient of variation high (>23%). Once adjusted for the standard length, the mean value of K was 2.65 ± 0.05 (between 3.26 for *Andinoacara rivulatus* and 1.57 for *Leporinus ecuadorensis*), again showing highly significant differences (*p* > 0.001) between species (Appendix A).

## 4. Discussion

In general, the environment is known to be a key factor in the morphological variation of fish [51,52]. Some environmental conditions, expression of genetic differences and/or gene pool pauperization could be responsible for changes in morphology [53]. In many cases, the morphological changes are the result of an interaction between environmental factors and genetic plasticity [54,55,56], although it is not always easy to explain the causes of the morphological differences between populations [57]. In previous papers, factors such as production system (wild and cultured) and capture area (river vs. basin), among others, were studied to explain morphological variations in each native species [31,33,35].

The quantification of specific characteristics of an individual or group of individuals can show the degree of speciation induced by biotic and abiotic conditions and contribute to the definition of different species [58]. In this research, independently of these factors, homogeneity within each species and a strong distance between species were found (Figure 1). The phenotypic differences confirmed the existence of six morphologically differentiated species in the basin Guayas (CHG). However, analyzing variable by variable, similarities are observed between some species in some of them, although the ANOVA test for the species effect resulted in highly significant differences for all the traits. These morphometric differences are presumably due more to a genetic component than to environmental factors, since the fishes were caught in the fluvial system of CHG, where each species has responded in a different way to the modification of the habitat [59]. All of this is in line with that shown by Langerhans et al. [46], who stated that the different morphological patterns could be produced through genetic differences or phenotypic plasticity, since populations can diverge through alternative morphologies based on genetics or through phenotypes induced by the environment.

Sex did not significantly affect the morphometric characteristics of each of the species, despite the macroscopic differences between males and females. This agrees with previous works in *Cichlasoma festae* [31], *Clarias gariepinus* [41] and *Tor putitora* [60]. On the opposite, in *Andinoacara rivulatus* [33], *Dormitator latifrons* [35] and *Hoplias microlepis* [59], differences by sex were found. In this sense, each native species develops its own strategy of determination and sexual differentiation in comparison to other investigations, more oriented to achieve reproduction under controlled conditions.

The upper mean values detected in a large part of the morphometric variables of *Hoplias microlepis* could be the result of its genetic attributes [42]. In this sense, *Hoplias microlepis* coexists with *Oreochromis* spp. in the same place of the basin Guayas, due to its strength and defensive capacity. On the contrary, species such as the *Andinoacara rivulatus* moves to marginal areas that the tilapia avoids. The size of the fishes would also be conditioned by the predation pressure and the habitat conditions, since both influence growth and survival, as well as by overfishing that could be responsible for the apparently small body structure as there is a tendency to catch large fishes [42]. This fact could be reflected in species such as *Cichlasoma festae, Dormitator latifrons, Bryncon dentex* and *Leporinus ecuadorensis*, since the mean and maximum values of the total or standard length were lower than those obtained in previous works [6,61,62,63,64], most of which were done in the last decade. However, the data obtained in *Hoplias microlepis* or *Andinoacara rivulatus* were superior to those obtained in previous works [65,66]; this probably was because we used a larger sample size and also relied on experienced local fishermen.

The results obtained allowed to establish the morphometric differentiation in six Ecuadorian fish native species using nine measurements. The species analyzed could be discriminated by the morphometric models generated, therefore showing that discriminant analysis was useful for differentiating species. Besides this, these morphological variables could be used to increase the consistency of specimens’ classification in each species [42].

The high correct assignment rates (98.97%) are in line with that obtained by Turan et al. [67], Yakubu and Okunserbor [42] or Khan et al. [68], which once again highlights the usefulness of using this type of analysis for the morphometric characterization and differentiation of species. The errors produced in *Bryncon dentex* with *Hoplias microlepis* (5 errors) and *Leporinus ecuadorensis* (1 error) could be due to the fact that the three fishes have an elongated morphostructural model (Figure 2). However, the error produced between *Hoplias microlepis* and *Andinoacara rivulatus* is not very well known. It could be due to the fact that both fishes present different morphometric characteristics.

The Mahanalobis distances show the closeness or distance of the six species studied based on the morphostructural model. Despite the fact that visually the two ”vieja” fishes (red and blue) are more similar, the discriminant analysis shows that the morphostructural model of the *Andinoacara rivulatus* is more similar to that of the *Dormitator latifrons* than to that of the *Cichlasoma festae* one. On the other hand, *Bryncon dentex* specimens turned out to be the furthest from the morphostructural point of view, which may be due to the fact that it has a more elongated structure, a fact that is reflected in the mean values of the adjusted variables, being the species with the highest mean values in variables related to length (total length = 25.27; Pre-dorsal fin length = 14.90; Pre-pelvic fin length = 11.72; Pre-anal fin length = 18.36; anal fin length = 5.27). The grouping of species in a cluster obtained by Cavalcanti et al. [40], where the species of the same genus were grouped in the same cluster, cannot be applied in our case, as we do not have two species of the same genus.

In our case, the groupings were made by morphological similarity, and, probably, not considered factors are involved, such as habitat preference, food, determination, sexual maturity, etc. The clusters obtained do not agree with Cavalcanti et al. [40], as they obtained a clear separation of species with a small body from those with a large body. The low overlap among specimens of each species is related with strong homogeneity obtained within each species for the morphological variables (i.e., *Leporinus ecuadorensis*, *Bryncon dentex* and *Andinoacara rivulatus*).

The K factor is an index used for monitoring feeding intensity, age and growth rates of fishes [69], and it is strongly influenced by environmental conditions and can be used as an index to evaluate the state of the aquatic ecosystem in which the fishes live [31]. However, the interpretation of this factor must be done carefully, since it may depend on several factors [70,71] such as food availability [72] or seasonality [73,74], including their interrelationship [72].

## 5. Conclusions

The aim of this research was to analyze morphometric differentiation among six Ecuadorian fish native species in the basin Guayas (CHG) and to demonstrate the usefulness of discriminant analysis. The six species analyzed could be discriminated by the morphometric model generated, therefore showing that discriminant analysis was useful for differentiating species. Nine biometric variables had the highest discriminant power and were sufficient for species discrimination, these variables being: pre-dorsal fin length, pre-orbital length, pectoral fin length, body depth 1, pelvic fin length, body depth 2, body width 2, body perimeter 3 and body width 1. Discriminant analysis methodology showed significant differences between species, as well as uniformity within each species. This methodology is direct, simple and economic. Therefore, it favors application in situ in rural communities, in developing countries and it could also be useful for the implementation of a livestock development program.

More evidence drawn from biochemical and molecular genetics will also consolidate the information derived from morphological variability. However, these will be the subjects of future research.

## Figures and Tables

**Figure 1 animals-11-00111-f001:**
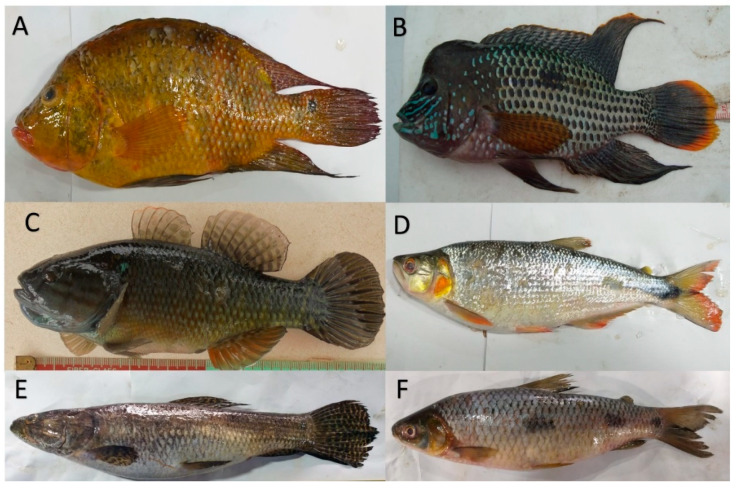
Six native freshwater species used in this study. (**A**) Cichlasoma festae; (**B**) Andinoacara rivulatus; (**C**) Dormitator latifrons; (**D**) Brycon dentex; (**E**) Hoplias microlepis; (**F**) Leporinus ecuadorensis.

**Figure 2 animals-11-00111-f002:**
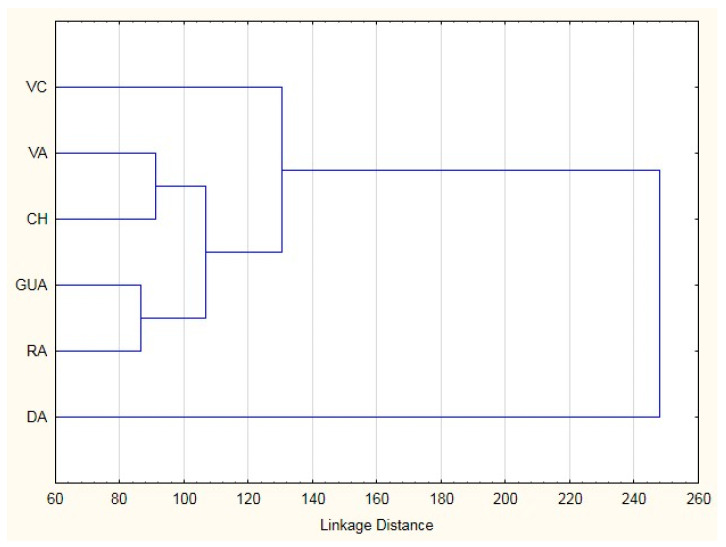
Cluster from Mahalanobis distances in six freshwater species from Ecuador. VC: *Cichlasoma festae*; VA: *Andinoacara rivulatus*; CH: *Dormitator latifrons*; GUA: *Hoplias microlepis*; RA: *Leporinus ecuadorensis*; DA: *Brycon dentex*.

**Figure 3 animals-11-00111-f003:**
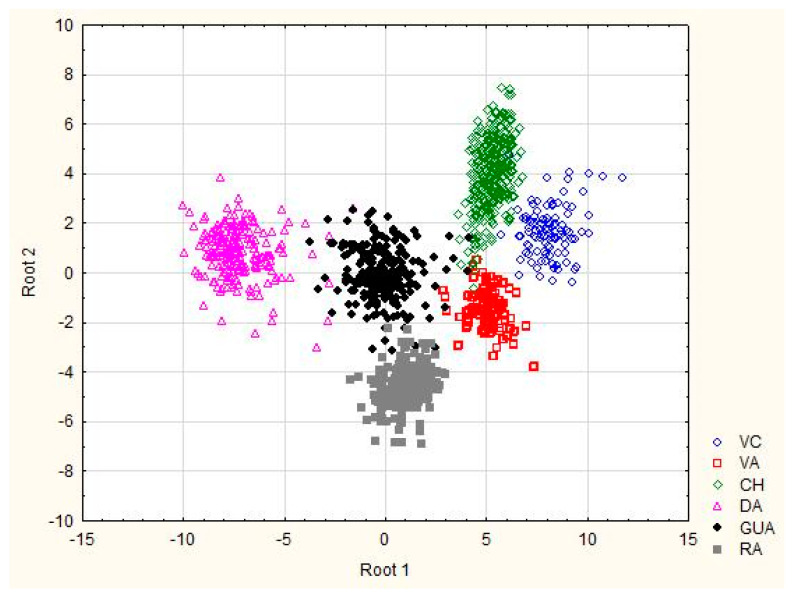
Plot of the individual observation discriminant scores obtained with the canonical discriminant function for six freshwater species from Ecuador. VC: *Cichlasoma festae*; VA: *Andinoacara rivulatus*; CH: *Dormitator latifrons*; GUA: *Hoplias microlepis*; RA: *Leporinus ecuadorensis*; DA: *Brycon dentex*.

**Table 1 animals-11-00111-t001:** Morphometric measurements used in this study.

Character	Description	Acronyms
Weight	Total weight including gut and gonads	BW
Total length	Tip of the upper jaw to the bottom of caudal end of the caudal fin	TL
Total length 1	Tip of the upper jaw to the top of caudal superior end of the caudal fin	TL 1
Total length 2	Tip of the upper jaw to the top of caudal inferior end of the caudal fin	TL 2
Standard length	Tip of the upper jaw to the tail base	SL
Head length.	From the front of the upper lip to the posterior end of the opercula membrane	HL
Eye diameter	The greatest bony diameter of the orbit	ED
Pre-orbital length	Front of the upper lip to cranial eye edge	Pre-OL
Pre-dorsal fin length	Front of the upper lip to the origin of the dorsal fin	Pre-DL
Pre-pectoral fin length	Front of the upper lip to the origin of the pectoral fin	Pre-PcL
Pre-pelvic fin length	Front of the upper lip to the origin of the pelvic fin	Pre-PvL
Pre-anal fin length	Front of the upper lip to the origin of the anal fin	Pre-AL
Dorsal fin length	From base of first dorsal spine to base of last dorsal ray	DFL
Dorsal fin ray length	From base to tip of the fifth dorsal ray	DFRL
Pectoral fin length	From base to tip of the pectoral fin	PcFL
Pelvic fin length	From base to tip of the pelvic fin	PvFL
Anal fin length	From base of first anal spine to base of last anal ray	AFL
Anal fin ray length	From base to tip of the last anal ray	AFRL
Upper jaw length	Straight line measurement between the snout tip and posterior edge of maxilla	UJL
Body perimeter 1	Body perimeter at the level of the first ray of the dorsal fin	P1
Body perimeter 2	Body perimeter at the level of the first radius of the anal fin	P2
Body perimeter 3	Body perimeter at the level of the last ray of the dorsal fin	P3
Body width 1	Straight line measurement from side to side at the level of the base of first dorsal spine	LC1
Body width 2	Straight line measurement from side to side at the level of the base of first anal spine	LC2
Body width 3	Straight line measurement from side to side at the level of the base of last dorsal ray	LC3
Body depth 1	Body depth at the level of the first ray of the dorsal fin	AC1
Body depth 2	Body depth at the level of the first ray of the anal fin	AC2
Body depth 3	Body depth at the level of the first radius of the caudal fin	AC3

**Table 2 animals-11-00111-t002:** Descriptive statistics of body measurements (original data) of six freshwater species from Ecuador (Mean ± Standard Error (Coefficient of Variation)).

Character ^1^	*Cichlasoma festae*	*Andinoacara rivulatus*	*Dormitator latifrons*	*Brycon dentex*	*Hoplias microlepis*	*Leporinus ecuadorensis*	ANOVA ^2^
BW	89.20 ^b^ ± 3.01 (23.89)	154.71 ^ac^ ± 2.58 (16.98)	173.13 ^a^ ± 5.37 (31.00)	150.91 ^a^ ± 5.47 (53.02)	442.66 ^d^ ± 29.52 (68.97)	106.10 ^bc^ ± 3.91 (38.30)	98.48 ***
K	3.17 ^b^ ± 0.11 (23.72)	5.40 ^d^ ± 0.13 (24.15)	2.41 ^b^ ± 0.06 (23.23)	4.27 ^c^ ± 0.16 (56.26)	1.47 ^a^ ± 0.10 (69.70)	1.66 ^a^ ± 0.07 (41.29)	115.55 ***
TL	18.30 ^a^ ± 0.27 (10.62)	18.37 ^a^ ± 0.15 (8.49)	24.65 ^c^ ± 0.32 (12.93)	21.39 ^b^ ± 0.23 (15.68)	37.82 ^d^ ± 0.69 (18.97)	20.82 ^b^ ± 0.29 (14.35)	368.014 ***
SL	14.37 ^ab^ ± 0.36 (17.74)	14.34 ^a^ ± 0.13 (9.32)	19.23 ^c^ ± 0.25 (13.24)	15.58 ^b^ ± 0.16 (14.82)	32.14 ^d^ ± 0.60 (19.28)	18.88 ^c^ ± 0.27 (15.00)	456.49 ***
HL	5.33 ^ab^ ± 0.08 (10.72)	4.88 ^a^ ± 0.04 (9.16)	6.47 ^d^ ± 0.10 (15.88)	5.62 ^b^ ± 0.06 (16.34)	8.31 ^e^ ± 0.15 (18.77)	3.88 ^c^ ± 0.06 (17.15)	264.61 ***
ED	1.21 ^bc^ ± 0.02 (10.69)	1.09 ^b^ ± 0.02 (15.43)	0.88 ^a^ ± 0.02 (18.86)	1.29 ^cd^ ± 0.02 (22.21)	1.34 ^d^ ± 0.03 (23.71)	0.79 ^a^ ± 0.02 (23.72)	104.12 ***
Pre-OL	2.20 ^c^ ± 0.08 (24.47)	2.12 ^c^ ± 0.03 (15.42)	1.21 ^a^ ± 0.02 (20.23)	1.18 ^a^ ± 0.02 (19.84)	1.34 ^b^ ± 0.04 (31.79)	1.41 ^b^ ± 0.02 (17.99)	193.28 ***
Pre-DL	5.37 ^b^ ± 0.11 (14.66)	6.31 ^c^ ± 0.05 (8.80)	7.68 ^a^ ± 0.11 (13.88)	12.76 ^d^ ± 0.13 (15.40)	16.16 ^e^ ± 0.30 (19.25)	8.32 ^a^ ± 0.13 (16.15)	546.75 ***
Pre-PcL	5.60 ^a^ ± 0.07 (9.25)	5.65 ^a^ ± 0.04 (8.01)	6.45 ^c^ ± 0.09 (13.77)	5.85 ^a^ ± 0.07 (18.18)	8.33 ^d^ ± 0.17 (20.79)	4.38 ^b^ ± 0.06 (15.26)	170.76 ***
Pre-PvL	5.82 ^a^ ± 0.09 (10.79)	6.26 ^a^ ± 0.06 (9.91)	3.36 ^c^ ± 0.12 (35.12)	11.18 ^b^ ± 0.12 (15.16)	11.08 ^b^ ± 0.22 (20.63)	8.56 ^d^ ± 0.19 (22.84)	441.05 ***
Pre-AL	9.13 ^a^ ± 0.16 (12.51)	10.39 ^a^ ± 0.11 (10.83)	12.95 ^b^ ± 0.19 (14.67)	15.35 ^c^ ± 0.15 (14.38)	25.29 ^d^ ± 0.55 (22.44)	13.12 ^b^ ± 0.21 (16.87)	393.80 ***
DFL	5.94 ^d^ ± 0.11 (13.39)	8.79 ^e^ ± 0.08 (9.60)	3.08 ^b^ ± 0.05 (15.26)	2.52 ^a^ ± 0.04 (22.26)	3.58 ^c^ ± 0.08 (22.28)	2.47 ^a^ ± 0.04 (16.17)	1,656.95 ***
DFRL	5.87 ^e^ ± 0.07 (8.94)	1.02 ^b^ ± 0.04 (40.37)	2.22 ^c^ ± 0.04 (18.50)	3.49 ^a^ ± 0.05 (20.13)	3.99 ^d^ ± 0.08 (20.37)	3.59 ^a^ ± 0.05 (14.59)	564.22 ***
PcFL	8.06 ^e^ ± 0.15 (13.42)	4.63 ^b^ ± 0.06 (12.95)	4.51 ^ab^ ± 0.07 (15.45)	4.34 ^a^ ± 0.05 (17.89)	2.80^d^±0.08 (28.85)	2.25 ^c^ ± 0.06 (25.58)	501.60 ***
PvFL	6.41 ^e^ ± 0.07 (8.26)	4.63 ^d^ ± 0.06 (13.10)	4.01 ^c^ ± 0.07 (17.86)	2.93 ^a^ ± 0.04 (17.91)	3.63 ^b^ ± 0.09 (25.49)	2.78 ^a^ ± 0.04 (15.78)	340.61 ***
AFL	4.59 ^c^ ± 0.18 (27.08)	3.08 ^a^ ± 0.03 (10.38)	2.75 ^a^ ± 0.05 (19.09)	5.64 ^d^ ± 0.09 (23.74)	2.99 ^a^ ± 0.08 (28.73)	1.82 ^b^ ± 0.04 (22.07)	323.08 ***
AFRL	3.30 ^b^ ± 0.07 (14.49)	3.46 ^b^ ± 0.06 (18.78)	2.03 ^c^ ± 0.07 (35.80)	2.51 ^a^ ± 0.04 (24.33)	2.95^d^±0.07 (24.30)	2.37 ^a^ ± 0.03 (14.09)	79.39 ***
UJL	0.94 ^a^ ± 0.02 (16.63)	0.64 ^b^ ± 0.01 (20.53)	1.12 ^d^ ± 0.02 (20.18)	1.01 ^a^ ± 0.02 (27.61)	2.18 ^e^ ± 0.05 (24.51)	0.80 ^c^ ± 0.02 (22.73)	371.62 ***
AC1	5.47 ^e^ ± 0.07 (9.39)	7.78 ^a^ ± 0.16 (20.62)	6.45 ^d^ ± 0.12 (17.92)	5.89 ^a^ ± 0.05 (12.00)	3.56 ^b^ ± 0.05 (14.14)	4.26 ^c^ ± 0.10 (24.78)	250.56 ***
AC2	4.94 ^ab^ ± 0.07 (9.68)	7.07 ^d^ ± 0.16 (23.55)	5.35 ^bc^ ± 0.08 (14.81)	5.44 ^c^ ± 0.05 (12.72)	4.77 ^a^ ± 0.06 (12.22)	4.91 ^a^ ± 0.11 (22.57)	79.12 ***
AC3	1.94 ^a^ ± 0.04 (15.55)	2.62 ^bc^ ± 0.05 (18.07)	2.93 ^c^ ± 0.04 (14.14)	2.36 ^ab^ ± 0.10 (64.39)	5.44 ^d^ ± 0.06 (11.69)	2.38 ^ab^ ± 0.05 (20.71)	1,87.981 ***
P1	13.28 ^a^ ± 0.16 (8.77)	16.28 ^d^ ± 0.10 (6.45)	14.85 ^c^ ± 0.20 (13.63)	13.61 ^a^ ± 0.18 (19.56)	13.66 ^a^ ± 0.19 (14.23)	10.62 ^b^ ± 0.13 (12.23)	93.92 ***
P2	11.38 ^a^ ± 0.11 (6.97)	14.99 ^c^ ± 0.11 (7.48)	12.23 ^ab^ ± 0.18 (14.58)	14.02 ^bc^ ± 0.62 (64.89)	14.19^bc^ ± 0.24 (17.69)	11.99 ^a^ ± 0.13 (11.32)	6.90 ***
P3	4.71 ^b^ ± 0.08 (11.69)	6.02 ^a^ ± 0.05 (8.05)	6.79 ^c^ ± 0.08 (11.97)	5.84 ^a^ ± 0.06 (15.62)	12.36 ^e^ ± 0.13 (10.91)	8.93 ^d^ ± 0.12 (14.10)	207.44 ***
LC1	2.31 ^a^ ± 0.05 (13.95)	2.61 ^b^ ± 0.02 (9.09)	3.29 ^d^ ± 0.06 (17.75)	2.62 ^b^ ± 0.03 (17.29)	3.11 ^c^ ± 0.04 (12.55)	2.12 ^a^ ± 0.03 (14.27)	117.53 ***
LC2	1.55 ^a^ ± 0.06 (28.76)	1.83 ^b^ ± 0.02 (12.25)	2.89 ^e^ ± 0.04 (13.95)	2.71 ^d^ ± 0.04 (21.28)	3.13 ^f^ ± 0.04 (13.52)	2.31 ^c^ ± 0.03 (14.30)	165.87 ***
LC3	4.71 ^a^ ± 0.08 (11.69)	0.91 ^a^ ± 0.02 (18.99)	2.10 ^d^ ± 0.03 (16.16)	2.46 ^b^ ± 0.04 (21.92)	2.54 ^b^ ± 0.04 (17.36)	1.33 ^c^ ± 0.02 (15.10)	262.33 ***

^1^ BW = body weight; K = Fulton´s factor; TL = total length; SL = standard length; HL = head length; ED = eye diameter; Pre-OL = pre-orbital length; Pre-DL = pre-dorsal fin length; Pre-PcL = pre-pectoral fin length; Pre-PvL = pre-pelvic fin length; Pre-AL = pre-anal fin length; DFL = dorsal fin length; DFRL = dorsal fin ray length; PcFL = pectoral fin length; PvFL = pelvic fin length; AFL = anal fin length; AFRL = anal fin ray length; UJL = upper jaw length; AC1 = body depth 1; AC2 = body depth 2; AC3 = body depth 3; P1 = body perimeter 1; P2 = body perimeter 2; P3 = body perimeter 3; LC1 = body width 1; LC2 = body width 2; LC3 = body width 3. ^2^ *** *p* < 0.001. ^a, b, c, d, e, f^ superscript letters indicate significative differences amongst species (*p* < 0.05).

**Table 3 animals-11-00111-t003:** Discriminant functions for the morphometric variables (fitted data) of six freshwater species from Ecuador.

Character ^1^	Wilks’-Lambda	Partial-Lambda	F-Remove	*p*-Level ^2^	Toler	1-Toler
Pre-DL	0.00	0.34	255.47	***	0.62	0.38
HL	0.00	0.82	29.05	***	0.54	0.46
Pre-OL	0.00	0.67	63.81	***	0.66	0.34
PcFL	0.00	0.75	43.36	***	0.47	0.53
AC1	0.00	0.73	49.13	***	0.17	0.83
Pre-PvL	0.00	0.80	32.27	***	0.70	0.30
PvFL	0.00	0.75	44.74	***	0.53	0.47
AFRL	0.00	0.82	28.76	***	0.66	0.34
AC2	0.00	0.84	25.11	***	0.23	0.77
ED	0.00	0.87	19.20	***	0.78	0.22
P1	0.00	0.87	19.06	***	0.64	0.36
AFL	0.00	0.90	14.56	***	0.71	0.29
UJL	0.00	0.87	19.91	***	0.84	0.16
LC2	0.00	0.92	12.09	***	0.36	0.64
Pre-AL	0.00	0.92	12.06	***	0.66	0.34
P3	0.00	0.90	14.80	***	0.52	0.48
LC1	0.00	0.92	11.64	***	0.36	0.64
TL	0.00	0.92	11.22	***	0.75	0.25
Pre-PcL	0.00	0.92	11.80	***	0.62	0.38
AC3	0.00	0.95	7.33	***	0.78	0.22
P2	0.00	0.99	1.79	Ns	0.73	0.27

^1^ TL = total length; HL = head length; ED = eye diameter; Pre-OL = pre-orbital length; Pre-DL = pre-dorsal fin length; Pre-PcL = pre-pectoral fin length; Pre-PvL = pre-pelvic fin length; Pre-AL = pre-anal fin length; PcFL = pectoral fin length; PvFL = pelvic fin length; AFL = anal fin length; AFRL = anal fin ray length; UJL = upper jaw length; AC1 = body depth 1; AC2 = body depth 2; AC3 = body depth 3; P1 = body perimeter 1; P2 = body perimeter 2; P3 = body perimeter 3; LC1 = body width 1; LC2 = body width 2. ^2^ *** *p* < 0.001; ns = not significantly different.

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
