# Peer review of "Usefulness of Discriminant Analysis in the Morphometric Differentiation of Six Native Freshwater Species from Ecuador"

_animals, 2021, doi:10.3390/ani11010111_

Round 1
Reviewer 1 Report
An interesting manuscript exploring morphometrics of fish of commercial important in the Guayas Hydrographic Basin. The study forms part of a larger body of work assessing morphological differentiation between and within species of fish. Overall, the study is well-designed and appears consistent with previous studies conducted by this research group. Most of the methodology has been very well justified. I have only a few minor suggestions and one major recommendation.
Generally the English is good. There were a couple of occasions where it was a little difficult to understand but I don’t think the message is lost.
Minor
Line 146: I think it's useful that the authors link to their previous paper ([25]) which had a diagram indicating all of the morphometric measurements. My suggestion isn't entirely necessary but I wonder if they could make this reference stand out a little more, for example: “A total of twenty-seven morphometric measurements were obtained, using 20 landmarks (Table 1, [25)), in agreement with the methodology used for native species of Ecuador [21,23,25].” (I have underlined the suggested change but other solutions could be applicable).
Line 158: Binomial names should be italicised. Here, the authors refer to Brycon alburnus but, earlier in the manuscript (Line 124), the authors state that they collected Brycon dentex. I don’t think they are synonymous so could the authors clarify?
Results: Where you have reported a P value from an ANOVA you need to provide the test statistic and degrees of freedom as well.
Lines 210-212: You state “The existence of different and unique morphostructural models for each species is observed in figure 3, which shows a clear spatial distribution of each fish species, with little overlap of certain individuals of some of them.” I think this should instead refer to figure 4?
Discussion: Much of the first paragraph on focuses on environment as a driver of morphological differences, but this was not a focus of the study. It was not even really mentioned in the Introduction. I think it could be incorporated into the rest of the discussion but the first paragraph needs to highlight what the study itself accomplished/explored. What were your findings? You only really start describing them in the second paragraph
You do not need to repeat P values or statistical analysis in the Discussion.
Lines 251-254: especially because references here are provided as numbers, rather than author/date, perhaps it would be helpful to quickly indicate when the previous works occurred. Secondly, have you any idea why the pattern in guandiche and vieja azul was different?
Major
In this study, the researchers euthanise study fish by immersion in water/ice mixture at 0.8 degrees C. I note that I have previously reviewed a manuscript from the same research group (Gonzalez-Martinez et al., 2020) where I made the same point as the below. Unfortunately I was unable to review the modifications made in response. However, I have been able to see the initial response from the authors, so thank you to the authors for their response. I also note that the authors have used the revised text in this current manuscript; thank you.
Nevertheless, I am still concerned about the information being provided here. The authors refer to the procedures outlined in the Canadian Council on Animal Care (CCAC, 2005) guide; however, the guide does not list immersion in ice as an acceptable procedure for the euthanasia of fish and, in fact, lists it as a method to be avoided. Furthermore, the guidance indicates that any physical methods of euthanasia should be followed by destruction of the brain to ensure death of the animal, a procedure not followed within this manuscript.
The authors also state that fish were stunned by immersion for 20 minutes, and then death certified. How was death certified?
I should highlight the relevant sections of the CCAC guidance:
[Guidelines on: the care and use of wildlife, I. Euthanasia]: “When properly carried out, stunning and pithing will produce rapid unconsciousness, but not death, and should only be used in combination with other techniques such as exsanguination. Likewise, physical methods such as freezing and drowning are unacceptable unless used in combination with anesthesia to ensure that the animal is deeply anesthetized at the time of euthanasia.”
[Guidelines on: the care and use of fish in research, teaching and testing, Guideline 113]: “Use of hypothermia (including putting fish on ice) before euthanasia should be avoided.”
The study used a large number of fish (n=1355), thus an acceptable method of euthanasia would have been use of immersion anaesthetic at lethal dose.
One of my main concerns here is that the authors are in danger of misrepresenting the guidance provided by the CCAC, and, whilst it is obviously too late for the fish utilise din the study, it needs to be clear that the protocol used is not appropriate under current guidelines and should not be used in future studies.
I appreciate the changes the authors made in the previous publication in response to my review, and that they considered this to be a standard protocol for work with fish morphometrics. It is, however, critical that ethical treatment of animals in research is maintained at the highest possible standard. Whilst I am sure the authors did what they practically could in the conditions available, I cannot consider this publication appropriate without further justification and/or clear indication within the manuscript that the suggested method of euthanasia by the referenced authority (CCAC) was not used. It would be unethical to publish this study without some attempt to ensure readers – including researchers interested in the methods of the study – are aware of the appropriate procedures and guidelines for euthanasia in fish.
I am happy to discuss with the authors.
References
Gonzalez-Martinez, A.; Lopez, M.; Molero, H.M.; Rodriguez, J.; González, M.; Barba, C.; García, A. (2020) Morphometric and Meristic Characterization of Native Chame Fish (Dormitator latifrons) in Ecuador Using Multivariate Analysis. Animals, 10, 1805.
CCAC guidelines on: the care and use of wildlife (2010) Available online at: http://www.ccac.ca/Documents/Standards/Guidelines/Wildlife.pdf. Last accessed: 18/12/2020
CCAC guidelines on: the care and use of fish in research, teaching and testing, Guideline 113 (2005) Available online at: http://www.ccac.ca/Documents/Standards/Guidelines/Fish.pdf. Last accessed: 18/12/2020
Author Response
We would like to thank for your work and valuable comments that have substantially help us to improve the quality of our initial manuscript. We have considered your comments and we have done all the suggested corrections. To improve our work, we highlight the changes asked for you and rest of reviewers in bold letters. A letter for each reviewer has been uploaded where each question posed is answered
We hope that you like the new version of the manuscript.
Thank you very much for your attention.
Kind Regards,
The authors

Reviewer 2 Report
The aim of the paper was to analyze morphometric differentiation in six Ecuadorian fish species and to demonstrate the usefulness of discriminant analysis. The species analyzed could be discriminated by the morphometric models generated, therefore showing that discriminant analysis was useful for differentiating species. However, I am not an Ecuadorian fish expert, but analyzed species seem to be relatively easy to discriminate with normal morphological determination methods, at least it looks for me like this from the pictures presented. But maybe this is more difficult, but that has to be mentioned in the manuscript then and also maybe for which species it is difficult. Therefore, I think it should be stated much clearer throughout the paper, why it is important to conduct this quite time consuming analysis for these species. Further, especially abstract and introduction need to be linguistically revised. Further, in the abstract many details are given, which are difficult to understand without reading the manuscript and should be therefore rewritten to better and more general present the main findings and highlight much more why these analysis are important and useful for conservation. Further, are all measurements necessary to get the results or which are the most important measurements? Are there measurements which are not needed to get a similar precision of the models? This should be explored more in my opinion. Were specimens all collected at one site ore at different sites? This is not explained well in the paper. They were collected in different years and seasons, so in my opinion it would be important to test if there were differences among sites (if more than one site was sampled)/ year/season. Further, it is stated that non-representative specimens were not used, but how can you then be sure that your models really work? In my opinion they would need to be included to test the robustness of the models.
In the following, I will give more detailed comments to the different parts:
Simple summary:
Line 19: are endangered
Abstract:
As mentioned before, the abstract needs to be rewritten to better summarize and generalize the findings of the study. Further, some sentences need to be linguistically revised.
Line 26-28: In my opinion, the aim of the study should be made much clearer in this first sentence. Why is it important to have morphostructural models differentiating the different species? Is it otherwise not (reliably) possible to distinguish them? So, the usefulness should not only be that it is possible but also indicate here, why it is important.
Line 29: Second part of the sentence sounds strange, please rephrase.
Line 30: what are morphometric adjusted variables? Give examples. Before giving the results, describe better what was actually done, e.g. by naming for example the most important measurements and methods. All measurements had power for discrimination, but were all needed to create the model?
Line 30-33: Do not present so much details here, as they are very difficult to understand, if first only reading the abstract. Describe better what the importance of this result is. What can you deduce from this? Further, if one is not an Ecuadorian fish specialist, I am not sure if it is directly clear that you are talking about different species, when you state e.g. vieja azul and chame specimens were grouped, you could also think that these are localities, so better use species here, if this is not explained before. Further, is this clustering concurrent with phylogenetic relationships among the species?
Line 33f: “the fishes from dama” now really sounds like a location. Maybe use scientific names instead here, that would help reducing the confusion. Further, I would say it is the other way around, they have greater morphostructural differentiation and show therefore a greater separation.
Line34: Do not just refer to a plot here, but state which analysis is shown in the plot.
Line 35f: which biological information is gained by this analysis which can help in the development of conservation strategies?
Line37f: In my opinion, you did not test for this, so I would leave it out here, or the conclusion must be much better reasoned in the discussion.
Line39: But what can be achieved with this information? Are really all measurements needed?
Introduction:
Line 46-50: This sentence has to be rephrased, difficult to understand.
Line 79: What is traditional gear and what is the non-traditional gear used?
Line 94: State in more detail, how welfare of individuals can be assessed by morphometric analyses, what are the key parameters? If you develop morphometric models, how do they help here?
Line 97: sentence need to be rephrased, further, why did you choose specifically theses species? (Line 99, italicize Cichlasoma festae). And I am not sure if not English trivial names should be given here instead of Spanish ones?
Line 102: You state here, that it would be of great interest to analyze morphological variations among species to assess morphological adaptations, but did you do that?
Line 114: Do you want to differentiate populations within species with this (this has not been tested here?) or do you mean species instead of populations? In general, the aim has to be formulated better here.
Material and Methods:
Where exactly were specimens captured? Provide a map with localities and coordinates of sampling sites! Were they really from the whole basin or only from one site? At how many sites were the specimens collected? How many of which species in which year and in which season (provide supplementary table)? Are there differences between localities/years/seasons? This should be tested.
Line 119: “an annual rainfall of 2400 and 838mm” ïƒ are these the values for the two provinces Los Rios and Manabi? Were the sampling sites located in both regions?
Line 128: What is meant with network here?
Line 129: What were non-representative specimens? In my opinion it would be important to include them in the analysis as well to test the robustness of the model.
Figure 1: State which picture belongs to which species!! And before, explain why you chose exactly these six species.
Line 153/154: What is meant by a “sample”? Is this species? And why are they grouped a priori?
Line 157: should be: assess if the data had equal variances.
Line 158: italicize species names
Line 165: sample = species?, in general, define what you mean with sample?
Line 179: … and was therefore not considered … So you combined data from both sexes for the following analyses? Should be stated here.
Results:
Line 192 and Table 2: What exactly is meant with relationships here? And what is the meaning of the letters after numbers in table 2? This needs to be explained! How I know it, same letters mean, that there is no significant difference between them and if letters are different then these are statistically different. But then not all comparisons are statistically different? And for which comparison is the p-Value given? Only for the comparisons which are significantly different? Should be clarified. Further, what are the numbers in brackets? Original data? But why are the measurements then so much higher than the original data?
Line 199: comment concerning letters and p-Values important also for supplementary tables.
Line 200: Sentence should be rephrased. And it should be tested if all measurements are needed to get this precision. Further, there is not much overlap between clusters in Figure 3 (named Figure 4 in the manuscript, but there is no figure 2), so I would expect more errors? (also Figure 3 has to be renamed to figure 2)
Line 206: is this clustering in accordance to phylogenetic relationships between species?
Line 212: I think it would be important to describe the overlap in more detail.
Figure 3 (named Figure 4): what is the percentage explained by the axes?
Line 222: Which differences were significant? Also here, if I interpret letters correctly, not all were significantly different?
Discussion:
Line 238: But as they are described species, I would not say that it is revealed by the analysis, but morphological differences between species are confirmed or detected also by the models. And it should not be “populations” but species here in the sentence.
Line 239: Where were similarities, this has to be described in the results section!
And here again, I have questions concerning the p-value, if there were similarities, how can all differences be significant? Or is this meant when combining all variables? Should be stated more precisely here.
Line 241: Which differences? Within species or among species? And I cannot follow the argumentation why the genetic component should explain the differences. What is meant by “in similar areas as they present the same geographical location”? Were they sampled at different locations and did you test for differences between them? This argumentation needs to be presented better or better reasoned as in this form it is not convincing for me.
Line 247: Not sure how you come to this conclusion, should be explained in more detail. And also it would be important to test if there are differences between years and seasons, as this could also influence morphometric variables.
Line 254: Were the specimens to which you compare your specimens sampled at the same locations or where were they from?
Line 255: Did you test of specimens sampled at river and farms differed from each other? should also be done.
Line 258: “it was believed convenient” is not a sufficient justification. And you cannot first say differences are due to genetic components because areas were similar and then use this for a justification for not differentiating when analyzing specimens from different habitat. This is somehow a circular argument and therefore not convincing.
Line 265: But population structure was not analyzed at all? It was as far as I understood it only used for to distinguish species, but not populations within species.
Line 266: Is this identification otherwise difficult between these species? This should definitely be described in more detail e.g. in the introduction and here it would also be important if non-representative specimens would have been included, as I guess for them these models might help.
Line 267f. : How is the precision if standard morphological identification methods are used? Is it lower? And again morphometric differentiation of populations was not tested, but species differentiation.
Line 285f: And what does this mean? Is one clustering approach better than the other? Or what can we get from this comparison?
Line 292f: This should be definitely tested and not only mentioned at the end of the discussion!!
Conclusions:
Line 298f: Does it? It has to be explained better, what new information about the species were gained and how this could help conservation.
Line 300f: The justification for this is not convincing to me. Either leave it out or explain/justify better.
Line 302f: Does it? I cannot follow the reasoning here. Should be explained better. What can be achieved by using this method?
Line 304f: But are all measurements needed? Which are the most important ones?
Author Response

(The authors gave the same response as above.)

Reviewer 3 Report
Overview: The manuscript reports on the suitability of morphometric indices in the identification of fish species originating from the Guayas hydrographic basin in Ecuador. Overall, the manuscript appears of solid quality: The study design is logical and straightforward and the results support the conclusions of the authors well. The level of written English could be improved but does not hinder the understanding of the manuscript.
Major comments:
L82-83: I feel that the manuscript would be strengthened by describing in more details how morphometric indices are necessary for this knowledge. Can’t these fish species be identified based on their external appearance (as I suspect the authors did at the time of sampling described in section 2.1)? And, if these are so similar to each other, would it not be safer to use a molecular method?
Section 2.1: Why were these six species selected rather than any other? I suppose that fish members from other species were rejected. On line 129, on what basis where fish considered “non representative specimen” and what percentage of the sampled fish were discarded at that stage?
L179: It is interesting that sex had no impact, as this is often such an important factor in fish. I would be interested in seeing this discussed by the authors, including in regard of the sampling season and the fish life cycle.
Discussion: I would suggest that the authors described the six species studied in more details and discuss their phylogenetic relationships.
L255-263: I think that this issue should be addressed, especially considering that the authors already have the relevant data.
Minor comments:
L29: Please add the letter “s” at the end of the word “species” and remove the word “in” before “all morphometric”.
L30: Please replace “discrimination” with “discriminatory”.
L32-33: Please add the scientific names of these species.
L55: “Tm” should be corrected to the correct scientific unit.
L62: Please name a couple of the most significant such limitations.
L65-66: By “in the Pacific”, I guess you mean “on the Pacific side of the continent”.
Figure 3: Please ass the scientific names for the species and please, compare these results to a phylogenetic tree of these fish.
Author Response

(The authors gave the same response as above.)

Round 2
Reviewer 2 Report
The manuscript has been greatly improved, thank you for considering many of the comments. Not all comments/suggestions for analyses were followed, but it is ok for me this way. I think to analyze or test for within species differences according to sampling season or location could have improved the study even more, but it is also ok without.
I only have some more minor comments, included in the attached pdf.

Author Response
We would like to thank for your work and valuable comments that have substantially help us to improve the quality of our initial manuscript. We have considered your comments and we have done all the suggested corrections. To improve our work, we use Track Changes function in Microsoft Word.
We hope that you like the new version of the manuscript.
Thank you very much for your attention.
Kind Regards,
The authors
Answer: We have added the following at the of the sentence: “, and to select a reduced number of direct, simple and low-cost measurements to be applied in marginal communities.”
2/ Line 36: change “high significative” by “highly significant”
Answer: Correction done
Answer: Correction done
4/ Line 37: significant --> this is somehow redundant to sentence before. Maybe say While sex within species had no significant effect... differences among species were significant. And then go to further detail with the 27 variables
Answer: We have changed “Sex had no significant effect on the morphology, but the specie showed significative differences.” By “While sex within species had no significant effect on the morphology, differences among species were significant.”
Answer: We have remove “All the variables were included in the discriminant analysis model although “ for improve the understanding the sentence
6/ Line 174-175: But a sample is one species, or sampled population of each species?And I am also not sure about the word "collected" here in reagard to statistical analysis? Might be good to rephrase the sentence further.
Answer: This is more cleared in the metohology, due to this we have removed it in statistical analisys part.
7/ Line 175: groups
Answer: Correction done
Answer: We have rewritten the sentence as follows: “A direct method of selection of variables was used at P-value < 0.05.”
Answer: We have written in present tense
10/ Line 247: change “adscrition” by “assignment”
Answer: Correction done
11/ Line 271: factors
Answer: We have rewritten the sentence because there is only one K factor. The new sentence is: “The difference of K factor was highly significant (P < 0.001) especially among Andinoacara rivulatus and Bryncon dentex with other species, and the coefficient of variation high (> 23%).”
Answer: We have no tested neither the date and season of capture. Therefore this paragraph is not derived from our results. It is correct your comment, and the paraghraph was removed.
13/ Line 306: not italics
Answer: Correction done
14/ Line 317: not in italics
Answer: Correction done
Answer (314 line): We have added “this probably was because we used a larger sample size and also relied on experienced local fishermen. ” to improve the sentence.
16/ Line 326: species
Answer: Correction done
17/ Line 329: for
Answer: Correction done
Answer: Correction done
Answer: We have rewritten the sentece following your suggestions as follows; “9 biometric variables had the highest discriminant power and were sufficient for species discrimination being: pre-dorsal fin length, pre-orbital length, pectoral fin length, body depth 1, pelvic fin length, body depth 2, body width 2, body perimeter 3 and body width 1.”
